# DiffPath: Generating Road Network based Path with Latent Diffusion Model

## Abstract

With the increasing use of GPS technology, path has become essential for applications such as navigation, urban planning, and traffic optimization. However, obtaining real-world path presents challenges due to privacy concerns and the difficulty of collecting large datasets. Existing methods, including count-based and deep learning approaches, struggle with two main challenges: handling complex distributions of road segments and ensuring global coherence in generated paths. To address these, we introduce DiffPath, a path generation model based on Latent Diffusion Models (LDMs). By embedding path into a continuous latent space and leveraging a transformer architecture, DiffPath captures both local transitions and global dependencies, ensuring the generation of realistic paths. Experimental results demonstrate that our model outperforms existing approaches in generating paths that adhere to real-world road network structures while maintaining privacy.

## 1 Introduction

With the widespread adoption of GPS technology and mobile devices, path has become essential for optimizing navigation systems (Thomason et al., 2020), supporting smart city planning (Lin et al., 2020), and route planning (Bibri, 2021). However, the collection and utilization of path raise significant privacy concerns, as such data often contains sensitive information regarding individuals' movements (Lu et al., 2019; Monreale et al., 2023; Zhu et al., 2024a). Additionally, large-scale collection of this data faces challenges due to regulatory constraints, such as the General Data Protection Regulation (GDPR) in Europe.

Recent studies have highlighted synthetic data generation as a promising alternative that ensures privacy protection while preserving data utility (Long et al., 2023; Zhu et al., 2024a;b). Despite the potential of synthetic path to augment datasets and substitute real data in privacy-sensitive applications, generating synthetic data presents two significant challenges: 1) **capturing complex path distributions**, 2) **ensuring global coherence in generated paths**.

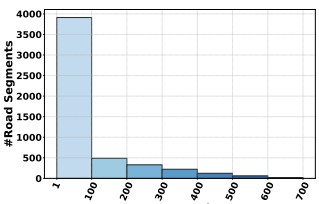

Figure 1: Statistics of Road Segment Traversals.

Path generation in urban road networks is particularly challenging due to the inherent inhomogeneity of the limited real-world dataset, which exhibits a clear long-tail distribution of road segment traversals. Figure 1 illustrates the statistics of road segment traversal in real world dataset (details provided in the 5.1.1), highlighting that some road segments, located in urban centers or transportation hubs, are frequently used and well-documented. Yet many road segments, particularly in more remote areas, are rarely traversed and thus suffer from limited data availability (Zhao et al., 2022; Zhu et al., 2022). This imbalance prevents the existing models from learning a comprehensive understanding of the complete road network in a city.

Existing path generation methods can be categorized into count-based methods and deep learning methods. The **count-based** methods generate paths by analyzing the frequency of transitions within historical path (Baratchi et al., 2014; Sutton et al., 1999). Due to the unevenness of limited real datasets, many road segments lack adequate historical observations, leading the model to make inaccurate or zero probability estimates and thus ignore these road segments during path generation. On the other hand, **deep learning methods** utilize neural networks to capture complex patterns in historical paths (Yin et al., 2017; Yu et al., 2017; Wang et al., 2022; Shi et al., 2024). Since

high-frequency paths appear more frequently in the dataset, the model can more easily fit to these common paths. However, low-frequency paths appear less in the data, and the model lacks sufficient training signals on them. As a result, the model struggles to generate paths that include these rare segments.

Another significant challenge in path generation for urban road networks is that the generated paths conform to the constraints of the road network, but are not realistic enough because they do not conform to most situations in reality. While ensuring connectivity between adjacent road segments is essential, models must also consider the long-range dependencies of non-adjacent road segments throughout the entire path to generate realistic paths (Yang et al., 2021). Ignoring the influence of non-adjacent road segments can lead to paths that are locally coherent but globally suboptimal or unrealistic. As illustrated in Figure 2, consider two paths from $v_1$ to $v_7$: $P_1 = [v_1, v_2, v_7]$ (depicted in blue) and $P_2 = [v_1, v_2, v_4, v_5, v_6, v_7]$ (depicted in red). Path $P_2$ traverses multiple intermediate nodes, resulting in a longer and unrealistic path. Although it is in line with the constraints of the road network, $P_2$ does not consider that selecting $v_4$ will result in a longer path to reach $v_7$, making the entire path unrealistic. Path $P_1$ represents the realistic path typically taken in most real-world scenarios. This example demonstrates how models focusing solely on adjacent road segments might generate unrealistic paths by failing to consider non-adjacent connections that lead to more optimal routes. Unfortunately, existing methods (Wang et al., 2022; Shi et al., 2024) neglect this critical issue.

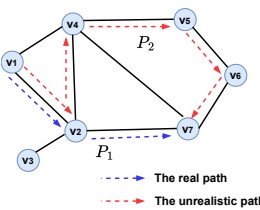

Figure 2: An example of real and unrealistic path.

To address the above challenges, we propose a method called **DiffPath** for path generation in urban road networks. **DiffPath** is designed to generate paths by effectively modeling complex data distributions and capturing long-range dependencies between road segments. Diffusion models have demonstrated flexibility in modeling complex distributions (Song et al., 2022; Zhu et al., 2024a; Shi et al., 2024). In **DiffPath**, we embed sequences of discrete paths into a continuous *latent space* where the diffusion process is applied. By iteratively denoising within the latent space, the model learns to reconstruct the original data distribution, including low-frequency paths. To further address the inherent bias toward high-frequency paths, we design a custom loss function for the diffusion model. Instead of predicting the added noise at each time step, the model directly predicts the initial latent representation.

Additionally, transformers are capable of modeling long-range dependencies through a self-attention mechanism (Peebles & Xie, 2023). To capture the correlations of both adjacent and non-adjacent road segments, we integrate a transformer-based architecture into the diffusion model. The self-attention mechanism enables the model to consider the long-range dependencies, ensuring global path consistency and realistic path generation. Moreover, We introduce a new similarity score specifically designed to rigorously evaluate path realism by considering both local transitions and global coherence. In summary, we make the following main contributions:

- We present DiffPath, the first latent diffusion-based approach for path generation, effectively modeling complex data distributions and capturing long-range dependencies between road segments.

- We introduce a custom loss function to address the long-tail distribution of road segments, ensuring diverse path generation. Additionally, we enhance diffusion with positional embeddings and a clamping mechanism for topological validity and contextual coherence.

- We validate DiffPath on two real-world datasets, showing it outperforms state-of-the-art models in generating high-fidelity, realistic paths while preserving data privacy.

## 2 RELATED WORK

### 2.1 PATH GENERATION FOR PATTERN MINING

Many methods have been developed to apply for path pattern mining. These approaches can broadly be categorized into two main categories: **count-based** and **deep learning-based models**.

Early models primarily relied on count-based methods, such as Markov chain models, which generate paths by capturing local state dependencies (Sutton et al., 1999). These models assume that the probability of transitioning to the next road segment depends solely on the current road segment, effectively modeling short-term dependencies. However, in the large state spaces typical of urban road networks, many state transitions are rare or absent from the training data, leading to sparsity issues. To mitigate this, hierarchical hidden states were introduced to reduce state space complexity and alleviate sparsity (Baratchi et al., 2014). Despite these improvements, count-based models still predominantly rely on information from the current and neighboring states, making it difficult to capture global dependencies and ensure long-range path consistency.

With advances in machine learning and reinforcement learning, deep learning-based models have become increasingly prominent in path generation. Generative Adversarial Networks (GANs) have been used to enhance path generation by integrating complex reward functions, improving both the diversity and quality of generated paths (Choi et al., 2021; Yu et al., 2017). However, GAN-based models often require large amounts of data and can struggle with the stability of training. Researchers have also increasingly turned to sequence-to-sequence (seq2seq) models for path generation (Wu et al., 2017). These models show promise for modeling long paths by encoding sequences of road segments and iteratively predicting subsequent road segments. However, adapting seq2seq models to respect the topological constraints of road networks presents a significant challenge. Architectures based on recurrent neural networks (RNNs) (Rao et al., 2020) have been explored to capture sequential dependencies, but they often suffer from the vanishing gradient problem when handling long sequences, limiting their ability to capture global path information.

To address some of these limitations, discrete diffusion models have been introduced (Shi et al., 2024). These models represent road network topology using adjacency matrices during the diffusion process, which is particularly effective for conditional task generation. However, in unconditional generation tasks, maintaining symmetry in the adjacency matrix can inaccurately represent one-way streets as bidirectional, introducing bias and inaccuracies in the generated paths.

## 2.2 DIFFUSION MODELS

The diffusion model, initially introduced as a probabilistic generation framework (Sohl-Dickstein et al., 2015), operates through two continuous processes: a forward process that gradually perturbs the data distribution by adding multiscale noise and a reverse process that reconstructs the data by learning its underlying distribution (Ho et al., 2020). Subsequent advances have significantly improved the quality of the generated samples and accelerated the sampling process. Further innovations include the introduction of a non-Markovian diffusion process (Song et al., 2021; Nichol et al., 2022), which reduces the number of sampling steps, and the approach of learning the variance in the reverse process to further streamline sampling. In addition, structural optimizations have been performed in reverse denoising neural networks to enhance the quality of the generated samples (Nichol & Dhariwal, 2021). As a cutting-edge generative model, the diffusion model has outperformed other models in various tasks, including computer vision (Rombach et al., 2022), natural language processing (Li et al., 2022), multimodal learning (Liu et al., 2023; Avrahami et al., 2022), and traffic prediction (Wen et al., 2024). Recent research has used diffusion models to generate synthetic data to improve datasets in specific domains (Zhu et al., 2024a;b), such as the generation of spatio-temporal trajectory data. Despite their widespread application, diffusion models have not been extensively explored for path generation. This is primarily due to the distinct nature of the task: path generation requires adherence to road network topology and sequential constraints (Wang et al., 2022; Shi et al., 2024), which are fundamentally different from the data structures handled in typical computer vision or natural language tasks.

## 3 PRELIMINARIES

### 3.1 PROBLEM DEFINITION

In this section, we introduce the definitions and notation that we use in this paper.

**Definition 1 (Road network)** *A road network is represented as a graph $G = (\mathbb{V}, \mathbb{E})$, where $\mathbb{V}$ is the set of vertices $v_i$, each representing a road intersection, and $\mathbb{E} \subseteq \mathbb{V} \times \mathbb{V}$ is the set of edges $e_i = (v_j, v_k)$, representing road segments from $v_j$ to $v_k$.*

**Definition 2 (Path)** *A path $p = \langle v_1, v_2, v_3, \ldots, v_L \rangle$ is a sequence of connected vertices, where each $v_i \in \mathbb{V}$ represents a vertex in the path, and two consecutive vertices are connected by an edge in $\mathbb{E}$.*

**Definition 3 (Path Generation)** *The objective is to develop a generative model capable of producing synthetic paths $\mathbb{N} = \{\hat{p}_1, \hat{p}_2, \ldots\}$, where the generated paths $\hat{p}_i$ preserve the structural and spatial characteristics of real-world paths.*

Table 1: Notations

| Notation | Description |
|---|---|
| $G = \langle \mathbb{V}, \mathbb{E} \rangle$ | A graph representing the road network with vertices $\mathbb{V}$ and edges $\mathbb{E}$. |
| $(v_0, v_1, \ldots, v_n)$ | The sequence of vertices in path $p$. |
| $e_i$ | The $i$-th road segment in path $p$. |
| $v_i$ | The $i$-th vertex in path $p$. |
| $p$ | A real-world path. |
| $\hat{p}$ | A generated synthetic path. |
| $z_t$ | The embedded path at time step $t$. |
| $\mathbb{R}$ | The real-world dataset. |
| $\mathbb{N}$ | The synthetic dataset. |
| $\alpha_t, \beta_t$ | Hyperparameters controlling the diffusion scale. |

## 4 METHODOLOGY

### 4.1 OVERVIEW

The model framework is illustrated in Figure 3. **DiffPath** begins by embedding the discrete path of length $l$ into a continuous latent space $\mathbb{R}^{l \times d}$ (referred to as **Encoding**). The embedded data then undergoes a forward and reverse diffusion process, allowing the model to effectively learn and represent complex patterns and dependencies within path in the latent space. The attention mechanism of the transformer enhances the model's ability to process sequential data and capture long-range patterns, enabling it to learn complex data distributions and generate new paths (referred to as the **Generation process**). Finally, the sampled results are mapped from the latent space back to discrete path (referred to as **Decoding**).

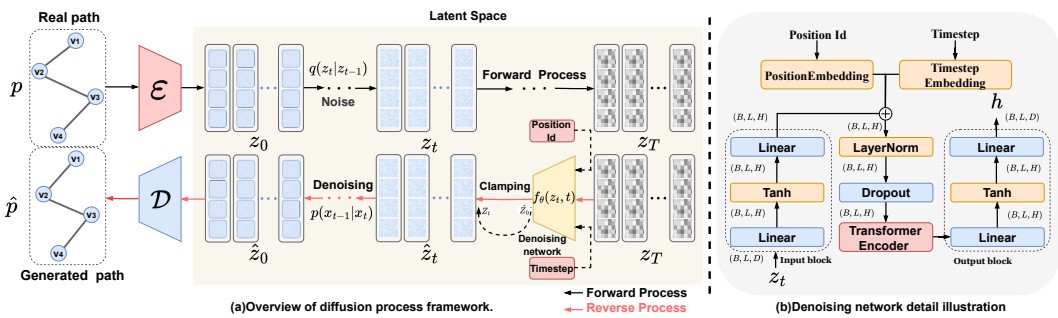

Figure 3: Overview of diffusion process framework. Discrete path is embedded into a continuous latent space and processed through a diffusion model to generate synthetic path.

### 4.2 ENCODING AND DECODING

To apply a latent diffusion model to the discrete path generation task, we first define an embedding function $E_{MB}(v_i)$ that maps each discrete path node $v_i$ (e.g., intersections or road segments) into a

vector in $\mathbb{R}^d$. For a path $\boldsymbol{p}$ of length $l$, the embedding of the path is represented as:

$$E_{MB}(\boldsymbol{p}) = [E_{MB}(v_1), E_{MB}(v_2), \ldots, E_{MB}(v_l)] \in \mathbb{R}^{l \times d}. \quad (1)$$

In order to operate within the latent space, we leverage a Markov transition in the forward process from the discrete path $\boldsymbol{p}$ to its latent continuous representation $\boldsymbol{z}_0$. This transition is parameterized as:

$$E_\phi(\boldsymbol{z}_0|\boldsymbol{p}) = \mathcal{N}(E_{MB}(\boldsymbol{p}), \sigma_0 I), \quad (2)$$

where $E_{MB}(\boldsymbol{p})$ is the embedded representation of the path, and $\sigma_0 I$ represents the covariance matrix of the Gaussian noise. By introducing noise into the latent space, the model captures both the underlying structure of the path and the inherent uncertainty in the transitions. Unlike pre-trained or fixed embeddings, we jointly learn both the path embeddings and the parameters of the diffusion model within the training objective. This end-to-end training enables the model to learn optimal latent representations suited to the path generation task.

During the reverse process, the continuous latent representation is gradually denoised and mapped back to the discrete path domain. This reverse diffusion process restores the path by iteratively refining the latent representation, using learned parameters to approximate the true distribution of the path. We select the most probable road segment at each position based on the current denoised representation $\boldsymbol{z}_0$. Formally, the mapping is defined as:

$$D_\theta(\boldsymbol{p}|\boldsymbol{z}_0) = \prod_{i=1}^{l} g_\theta(v_i|z_i), \quad (3)$$

where $g_\theta(v_i \mid z_i)$ is modeled by a softmax distribution over all possible road segments. The softmax distribution ensures that the probabilities of each possible road segment are normalized, and the road segment with the highest probability is selected. Rather than a single deterministic step, the reverse process involves a series of stochastic updates that gradually recover the discrete path from the noisy latent representation.

### 4.3 TRANSFORMER-BASED DIFFUSION PROCESS

**Forward and Reverse Process**  The forward process begins with the initial latent representation $\boldsymbol{z}_0$, progressively adding Gaussian noise over a series of steps, generating increasingly noisy states $\boldsymbol{z}_1, \boldsymbol{z}_2, \ldots, \boldsymbol{z}_T$. Each transition is defined by a Gaussian distribution:

$$q(\boldsymbol{z}_t|\boldsymbol{z}_{t-1}) = \mathcal{N}(\boldsymbol{z}_t; \sqrt{1 - \beta_t}\, \boldsymbol{z}_{t-1}, \beta_t I), \quad (4)$$

where $\beta_t$ represents the noise variance at time step $t$. By the final step $T$, the latent variable $\boldsymbol{z}_T$ becomes a near-Gaussian distribution.

The reverse process aims to reconstruct the original latent state $\boldsymbol{z}_0$ from the noisy state $\boldsymbol{z}_T$, using a learned denoising model. The reverse transitions follow the Gaussian distribution:

$$p_\theta(\boldsymbol{z}_{t-1}|\boldsymbol{z}_t) = \mathcal{N}(\boldsymbol{z}_{t-1}; \mu_\theta(\boldsymbol{z}_t, t), \sigma_\theta(\boldsymbol{z}_t, t)^2 I), \quad (5)$$

where $\mu_\theta(\boldsymbol{z}_t, t)$ and $\sigma_\theta(\boldsymbol{z}_t, t)^2$ are the predicted mean and variance, respectively, at each time step $t$, both of which are learned parameters.

**Training Objective**  Traditional diffusion models add noise at each step and learn to denoise the data in the reverse process to reconstruct the original data. This approach is effective for continuous data, where noise can be gradually removed due to the smooth nature of the data. This is typically done by minimizing a loss function designed to predict the added noise:

$$\min_\theta \mathbb{E}_{\boldsymbol{z}_t, \epsilon} \left[ \|\epsilon - \epsilon_\theta \left( \sqrt{\bar{\alpha}_t}\, \boldsymbol{z}_0 + \sqrt{1 - \bar{\alpha}_t}\, \epsilon, t \right) \|^2 \right], \quad (6)$$

where $\epsilon$ represents the noise added to the original data $\boldsymbol{z}_0$, and $\epsilon_\theta(\boldsymbol{z}_t, t)$ is the model's predicted noise. $\boldsymbol{z}_t$ represents the noisy data at time step $t$, and $\sqrt{\bar{\alpha}_t}$, $\sqrt{1 - \bar{\alpha}_t}$ are weighting coefficients of the diffusion process.

In path generation tasks, the data consists of discrete nodes (e.g., intersections or road segments) (Yang et al., 2023). Adding noise to such discrete data at each step can cause significant

deviations from the original structure, particularly for low-frequency road segments. Unlike continuous data, discrete data lacks a range of possible values, making noise introduction more likely to result in irreversible changes. Moreover, since the diffusion process involves iterative noise addition and removal, errors in predicting certain road segments, especially low-frequency ones, can accumulate over time. This accumulation can lead to the model ignoring or misrepresenting these road segments during path generation.

To overcome these limitations, we build on insights from previous work (Chen et al., 2023; Li et al., 2022) by directly predicting the latent representation $z_0$ for the entire path. Instead of progressively predicting noise at each step, the model learns the full path representation in one pass. This allows the model to capture sparse or uncommon road segments by focusing on the overall path structure, enhancing accuracy in path generation:

$$L_{train}(\boldsymbol{p}) = \sum_{t=1}^{T} \mathbb{E}_{q(\boldsymbol{z}_t|\boldsymbol{z}_0)} \|f_\theta(\boldsymbol{z}_t, t) - \boldsymbol{z}_0\|^2, \tag{7}$$

where $f_\theta(\boldsymbol{z}_t, t)$ is the model's prediction of $\boldsymbol{z}_0$ at time step $t$. By directly learning to predict $\boldsymbol{z}_0$. The predicted network structure is shown in the figure 3(b). We leverage transformer with its powerful self-focused mechanism to capture remote dependencies and sequential patterns in path. By combining time step information and location information, the model can well take into account local node transformation and global path dependence. To further enhance the model's performance during the reverse process, we apply the *clamping trick*, a method introduced in previous research (Li et al., 2022). In this method, the predicted latent state $f_\theta(\boldsymbol{z}_t, t)$ is clamped to the nearest valid road segment embedding, ensuring that each step in the reverse process adheres closely to the true discrete structure of the data. The clamping is performed as follows:

$$\boldsymbol{z}_{t-1} = \sqrt{\bar{\alpha}_t} \cdot \text{Clamp}(f_\theta(\boldsymbol{z}_t, t)) + \sqrt{1 - \bar{\alpha}_t}\epsilon, \tag{8}$$

where $\epsilon \sim \mathcal{N}(0, I)$, and the clamping operation ensures that the predicted vector aligns with a valid road segment in the embedding space. This approach reduces errors during the decoding process, improving the accuracy of the generated paths. The overall training objective combines the losses from the embedding process, diffusion process, and decoding process. This ensures that the model captures the full complexity of path while maintaining both local and global structure. The final loss is expressed as:

$$L_{train}(P) = \mathbb{E}_{q_\phi(\boldsymbol{z}_0|X)}\left[\sum_{t=1}^{T} \|f_\theta(\boldsymbol{z}_t, t) - \boldsymbol{z}_0\|^2 + \log E_\phi(\boldsymbol{z}_0|\boldsymbol{p}) - \log D_\theta(\boldsymbol{p}|\boldsymbol{z}_0)\right]. \tag{9}$$

The training process algorithm is as follows:

---

**Algorithm 1** Training Process of DiffPath

---

1: **Input:** Real path $\boldsymbol{p}$, number of diffusion steps $T$
2: **Output:** Trained model parameters $\theta$
3: **for** $i = 1, 2, \ldots$ **do**
4:     Embed $\boldsymbol{p}$ to continuous field
5:     Sample $t \sim \text{Uniform}(\{1, \ldots, T\})$, $\epsilon \sim \mathcal{N}(0, I)$, initialize $\boldsymbol{z}_0$
6:     Compute $\boldsymbol{z}_t = \sqrt{\alpha_t}\boldsymbol{z}_0 + \sqrt{1 - \alpha_t}\epsilon$
7:     Compute the predicted $\hat{\boldsymbol{z}}_0 = f_\theta(\boldsymbol{z}_t, t)$
8:     Compute loss $L_{train}(\boldsymbol{p})$
9:     Backpropagate and update model parameters $\theta$
10: **end for**

---

### 4.4 SAMPLING PROCESS

During the generation phase, the model begins with a noise vector $\boldsymbol{z}_T$ randomly sampled from a Gaussian distribution. The process iteratively applies the learned denoising function to the noise vector to generate realistic path. Specifically, at each time step $t$, the model uses the function $f_\theta(\boldsymbol{z}_t, t)$ to predict $\boldsymbol{z}_0$, and then samples of the distribution:

$$\boldsymbol{z}_{t-1} \sim q(\boldsymbol{z}_{t-1} \mid f_\theta(\boldsymbol{z}_t, t), \boldsymbol{z}_t) \tag{10}$$

to obtain the next intermediate state. This iterative process continues until the model reaches the initial state $z_0$. The final step is to decode $z_0$ back into the discrete path $\hat{p}$ using the process argmax $p_\theta(p \mid \hat{z}_0)$. This ensures that the generated path maps smoothly back into the discrete space, maintaining the structural and statistical properties of the original data.

---

**Algorithm 2** Generating Process of DiffPath

---

1: **Input:** Number of diffusion steps $T$, trained model parameters $\theta$
2: **Output:** Generated discrete path $\hat{p}$
3: Random sampling $z_T$
4: **for** $t = T, T - 1, \ldots, 1$ **do**
5:     Compute $f_\theta(z_t, t)$
6:     Sample $z_{t-1} \sim q(z_{t-1} \mid f_\theta(z_t, t), z_t)$
7: **end for**
8: Decode $\hat{z}_0$ to discrete path $\hat{p}$ using argmax $p_\theta(p \mid \hat{z}_0)$
9: **return** $\hat{p}$

---

## 5 EXPERIMENTS

In this section, we provide the basic experimental setup, comparison of the main results, and visualization analysis. We conduct extensive experiments on two real-world pathsets to demonstrate the superior performance of the proposed model in synthetic path datasets generation.

### 5.1 EXPERIMENTAL SETUPS

#### 5.1.1 DATASET

The raw datasets consist of GPS trajectories from two cities, referred to as Chengdu and Xi'an. The road network data for both cities was obtained from OpenStreetMap. Each road network is modeled as an directed graph. We then employed the map-matching algorithm proposed by (Meert & Verbeke, 2018) to align the GPS trajectory points to the road network. Consequently, the GPS trajectories were converted into path on the graph. For training, we randomly sampled 80% of the paths from each dataset, while 20% was set aside for testing. Please refer to the appendix B for more details

Table 2: Statistical Information of the Chengdu and Xi'an Datasets

| City | Number of Vertices | Number of Paths | Average Path Length |
|---|---|---|---|
| **Chengdu** | 2,865 | 91,070 | 24.737 |
| **Xi'an** | 2,675 | 63,110 | 24.921 |

#### 5.1.2 BASELINES.

In order to directly evaluate the effectiveness of our generation model, we perform a comparative analysis of several path generation algorithms. The **N-gram** model, which estimates the probability of transition $p(v_t|v_{t-1}, \ldots, v_{t-n+1})$ by counting the frequency of events, effectively captures local dependencies between paths. The Hidden Markov Model **(HMM)** (Yin et al., 2017) optimizes state reduction techniques to effectively simulate the underlying structure of path sequences. Additionally, the **MTNet** (Wang et al., 2022) model uses an architecture based on long-range memory (LSTM) to better capture long-range dependencies and complex sequential patterns in path. And the latest model **GDP** (Shi et al., 2024), which is modeled by the method of a discrete diffusion model. These comparisons provide a comprehensive assessment of the generative capabilities of our model relative to established methodologies.

### 5.1.3 EVALUATION METRICS

To evaluate the performance of our model, we employed several metrics to assess both local transitions and global structures in the generated paths. These metrics include **Kullback-Leibler Edge Visit Divergence (KLEV)**, **Jensen-Shannon Edge Visit Divergence (JSEV)** (Shi et al., 2024), and the newly introduced **Similarity Score (SS)**. **KLEV** measures the divergence between the edge visit frequency distributions of real and generated paths, capturing first-order transitions between adjacent nodes. It is defined as:

$$\text{KLEV}(P, P') = D_{\text{KL}} \left( \text{freq} \left( \forall (v_i, v_{i+1}) \in P \right) \| \text{freq} \left( \forall (v_i, v_{i+1}) \in P' \right) \right), \tag{11}$$

where freq$(\cdot)$ calculates the visiting frequencies of all edges $(v_i, v_{i+1})$ across the real paths $P$ and generated paths $P'$, and $D_{\text{KL}}$ denotes the Kullback-Leibler divergence. Similarly, **JSEV** provides a more stable and symmetric alternative by replacing the Kullback-Leibler divergence with Jensen-Shannon divergence. It is given by:

$$\text{JSEV}(P, P') = \text{JS} \left( \text{freq} \left( \forall (v_i, v_{i+1}) \in P \right) \| \text{freq} \left( \forall (v_i, v_{i+1}) \in P' \right) \right), \tag{12}$$

where $\text{JS}(p\|q)$ denotes the Jensen-Shannon divergence. Like KLEV, lower JSEV values indicate better alignment between real and generated path distributions.

While KLEV and JSEV capture first-order transitions, they do not account for multi-hop dependencies between non-adjacent road segments. To better assess the overall similarity between the generated and real paths, we define the **Similarity Score (SS)** as follows:

$$S(G, R_{\text{best}}) = \frac{|\mathbb{P}(G) \cap \mathbb{P}(R_{\text{best}})|}{|\mathbb{P}(G)|}, \tag{13}$$

$\mathbb{P}(G)$ represents the set of adjacent node pairs in the generated path $G$, and $R_{\text{best}}$ is the real path with the maximum overlap of adjacent node pairs with $G$. For a set of generated paths $\{G_1, G_2, \ldots, G_h\}$, we compute the similarity score $S(G_i, R_{\text{best}})$ for each generated path $G_i$ with its most similar real path $R_{\text{best}}$. The formula for the average similarity score across the entire set is given by:

$$S_{\text{avg}} = \frac{1}{h} \sum_{i=1}^{h} S(G_i, R_{\text{best}}), \tag{14}$$

where $S(G_i, R_{\text{best}})$ is the similarity score for the $i$-th generated path.

### 5.2 EXPERIMENTAL PERFORMANCE

### 5.2.1 OVERALL PERFORMANCE

Table 2 presents the performance comparison of our model and the selected baseline methods on two real-world datasets. We randomly generated 20,000 paths for each method and calculated all evaluation metrics. Our model outperforms other models based on neural networks in all metrics.

Table 3: Evaluations for Path generation.

| City | Metrics | N-gram | HMM | GDP | MTnet | DiffPath (Ours) |
|---|---|---|---|---|---|---|
| **Chengdu** | SS | 0.701 | 0.681 | 0.616 | 0.821 | **0.933** |
| | KLEV | 0.140 | 0.135 | 0.686 | 0.129 | **0.106** |
| | JSEV | 0.033 | 0.028 | 0.159 | 0.038 | **0.018** |
| **Xi'an** | SS | 0.628 | 0.633 | 0.571 | 0.772 | **0.893** |
| | KLEV | 0.133 | 0.130 | 0.697 | 0.127 | **0.122** |
| | JSEV | 0.031 | 0.025 | 0.147 | 0.033 | **0.023** |

We observe that for **KLEV** and **JSEV**, count-based models perform poorly as they depend on simple probability estimates for transitions between neighboring nodes. This limitation hinders their ability to capture rare or infrequent transitions, leading to lower performance on these metrics. Among the deep learning models, our approach achieves the lowest **KLEV** and **JSEV** values, indicating that the

distribution of the generated paths closely matches to the distribution of the real path distributions. All these observations demonstrate the effectiveness of **DiffPath.**

On **Similarity Score (SS)**, we can observe that **DiffPath** also achieves the SOTA compared to all other baselines. This shows that our model is especially good at generating paths that resemble real-world travel patterns. The transformer architecture plays a key role here by helping the model capture long-range dependencies across different road segments, ensuring that the generated paths reflect realistic travel routes over longer distances. In contrast, count-based models, which only focus on immediate transitions between adjacent nodes, struggle to represent the broader structure of paths. This limitation makes it difficult for these models to capture the full complexity of road networks, leading to lower SS values. As a result, while count-based models may produce paths that are locally correct, they fail to capture the overall travel patterns and structure seen in real-world data.

We also observed that the newly proposed **GDP** framework did not perform well on our dataset. This is primarily because GDP is designed for path planning tasks and focuses on conditional path generation. While it performs well in conditional scenarios, its unconditional path generation only ensures road network constraints without considering path distribution. Consequently, it performs poorly on metrics like KLEV and JSEV, which evaluate distributional characteristics.

### 5.2.2 VISUALIZATION ANALYSIS

We visualize the randomly selected real and generated paths in two cities. Figure 4 presents a visual comparison between the paths generated by **DiffPath** and the real-world paths for the cities of Chengdu and Xi'an. From these visualizations, it is evident that the generated paths align closely with the overall structure of the real paths, successfully capturing the major road networks in both cities. Key intersections and primary road segments are accurately represented in the generated paths, demonstrating the model's capacity to replicate real-world travel patterns.

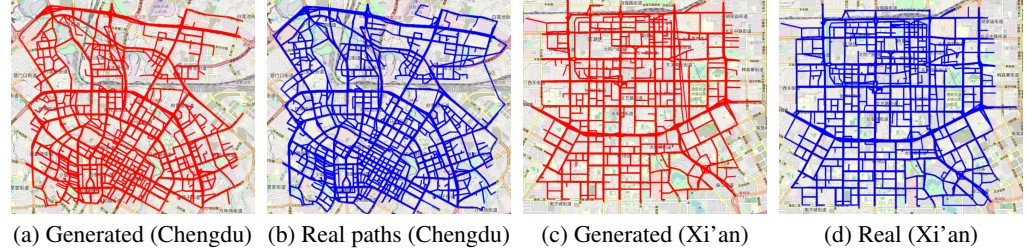

(a) Generated (Chengdu)   (b) Real paths (Chengdu)   (c) Generated (Xi'an)   (d) Real (Xi'an)

Figure 4: The visualization results of generated path and real path in Chengdu and Xi'an

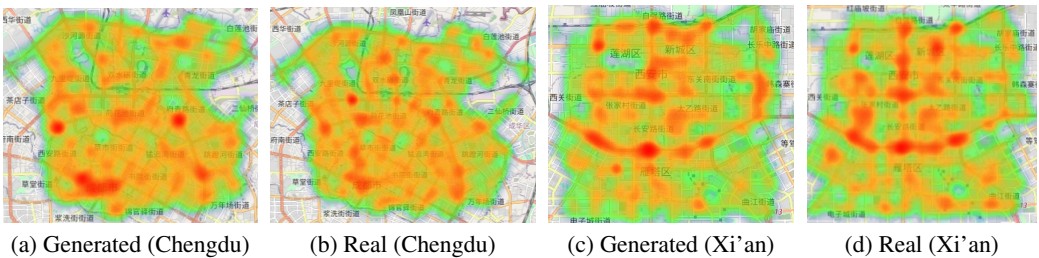

(a) Generated (Chengdu)   (b) Real (Chengdu)   (c) Generated (Xi'an)   (d) Real (Xi'an)

Figure 5: The visualized results of generated heat maps and real heat maps for Chengdu and Xi'an

To further evaluate the generated paths, Figure 5 compares the heatmaps of real and generated road segments for Chengdu and Xi'an. The heatmaps visualize the density and frequency of road segment usage, providing insights into the model's ability to capture the spatial distribution of traffic. In both cities, the generated heatmaps exhibit a distribution of high-traffic areas that closely mirrors the

actual heatmaps. This consistency indicates that the model effectively replicates high-frequency travel road segments observed in the real-world data, particularly in densely connected urban areas. More visualization results can be found in Appendix C.

### 5.2.3 ABLATION EXPERIMENT

For ablation experiment, we can see that the Transformer consistently performs better than UNet across all metrics in both cities from the results in Table 4. The higher Similarity Scores (SS) for the Transformer in Chengdu (0.933) and Xi'an (0.893) show that it captures the structure of the paths more accurately, both in terms of local connections and overall path flow. The lower KLEV and JSEV values for the Transformer also highlight its ability to model both common and rare road segments more effectively, addressing the issue of long-tail distributions. Overall, the Transformer aligns more closely with real-world paths, making it the better option for generating realistic paths in urban road networks compared to UNet.

### 5.2.4 PARAMETER EXPERIMENT

We observe that increasing the embedding dimension from 32 to 256 leads to significant improvements in all metrics. The results for both Chengdu and Xi'an are summarized in Table 5. For more detailed parameter settings, please refer to the appendix B

Table 4: Comparison of Transformer and UNet Architectures for Path Generation.

| City | Metrics | UNet | Transformer |
|---|---|---|---|
| **Chengdu** | SS | 0.915 | **0.933** |
| | KLEV | 0.122 | **0.106** |
| | JSEV | 0.023 | **0.018** |
| **Xi'an** | SS | 0.872 | **0.893** |
| | KLEV | 0.137 | **0.122** |
| | JSEV | 0.027 | **0.023** |

Table 5: Comparison of different embedding dimensions.

| City | Metrics | 32 | 64 | 128 | 256 |
|---|---|---|---|---|---|
| **Chengdu** | SS | 0.786 | 0.851 | **0.933** | 0.917 |
| | KLEV | 0.144 | 0.137 | **0.106** | 0.108 |
| | JSEV | 0.033 | 0.026 | **0.018** | 0.019 |
| **Xi'an** | SS | 0.708 | 0.816 | **0.893** | 0.887 |
| | KLEV | 0.144 | 0.137 | **0.122** | 0.125 |
| | JSEV | 0.033 | 0.026 | **0.023** | 0.023 |

## 6 CONCLUSION

In this work, we propose a path generation method based on a latent diffusion model (**DiffPath**). Our approach leverages the latent diffusion model's strength in generating discrete data and the transformer's ability to learn sequence features effectively. Specifically, real paths are first embedded into a latent space, where they undergo a forward noise trajectory process, gradually transforming them into random noise. The model then applies reverse trajectory denoising to reconstruct the paths from noise and finally decodes the synthesized paths. The effectiveness of **DiffPath** is validated through extensive experiments. Further experimental results show that the paths generated by the model align well with the statistical characteristics of real-world paths.

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

# A    TRAINING OBJECTIVE DERIVATION

In the context of continuous diffusion models, we start with the goal of minimizing the difference between the true posterior mean $\hat{\mu}(x_t, x_0)$ and the predicted mean $\mu_\theta(x_t, t)$ in each diffusion step. The objective function is represented as:

$$\|\hat{\mu}(x_t, x_0) - \mu_\theta(x_t, t)\|^2 . \tag{15}$$

Substituting the definitions of $\hat{\mu}(x_t, x_0)$ and $\mu_\theta(x_t, t)$, we have:

$$\hat{\mu}(x_t, x_0) = \frac{\sqrt{\bar{\alpha}_{t-1}}\beta_t}{1 - \bar{\alpha}_t} x_0 + \frac{\sqrt{\alpha_t}(1 - \bar{\alpha}_{t-1})}{1 - \bar{\alpha}_t} x_t,$$

$$\mu_\theta(x_t, t) = \frac{\sqrt{\bar{\alpha}_{t-1}}\beta_t}{1 - \bar{\alpha}_t} f_\theta(x_t, t) + \frac{\sqrt{\alpha_t}(1 - \bar{\alpha}_{t-1})}{1 - \bar{\alpha}_t} x_t.$$

Substituting these into the equation:

$$\|\hat{\mu}(x_t, x_0) - \mu_\theta(x_t, t)\|^2 \tag{16}$$

$$= \left\| \left( \frac{\sqrt{\bar{\alpha}_{t-1}}\beta_t}{1 - \bar{\alpha}_t} x_0 + \frac{\sqrt{\alpha_t}(1 - \bar{\alpha}_{t-1})}{1 - \bar{\alpha}_t} x_t \right) - \left( \frac{\sqrt{\bar{\alpha}_{t-1}}\beta_t}{1 - \bar{\alpha}_t} f_\theta(x_t, t) + \frac{\sqrt{\alpha_t}(1 - \bar{\alpha}_{t-1})}{1 - \bar{\alpha}_t} x_t \right) \right\|^2 \tag{17}$$

$$= \left\| \frac{\sqrt{\bar{\alpha}_{t-1}}\beta_t}{1 - \bar{\alpha}_t} (x_0 - f_\theta(x_t, t)) \right\|^2 . \tag{18}$$

Factoring in the constant $\frac{\sqrt{\bar{\alpha}_{t-1}}\beta_t}{1-\bar{\alpha}_t}$:

$$= \left( \frac{\sqrt{\bar{\alpha}_{t-1}}\beta_t}{1 - \bar{\alpha}_t} \right)^2 \|x_0 - f_\theta(x_t, t)\|^2 . \tag{19}$$

Since $\left( \frac{\sqrt{\bar{\alpha}_{t-1}}\beta_t}{1-\bar{\alpha}_t} \right)^2$ is a constant with respect to the parameters that are optimized, minimizing this expression is equivalent to the following.

$$\|x_0 - f_\theta(x_t, t)\|^2 . \tag{20}$$

This final form highlights that the model is trained to predict $x_0$ directly at each diffusion step, which simplifies the optimization process and improves stability.

# B    IMPLEMENTATION DETAIL

## B.1    CONFIGURATION

The paths in each dataset were selected based on recorded travel data, capturing real-world road usage across various regions of the cities. Each path consists of a sequence of connected vertices (road segments) representing actual vehicle routes.All experiments were implemented in PyTorch and conducted on a single NVIDIA GeForce RTX 3090 GPU.

## B.2    HYPERPARAMETERS

The hyperparameters specific to **DiffPath** include the number of diffusion steps, the embedding dimension, and the noise schedule. We set the number of diffusion steps to 2000 and the sequence length to 144. For the embedding dimension, taking into account computing resource consumption, we experiment with values in $d \in \{16, 64, 128, 256\}$, choosing $d = 128$ for both city datasets.

**DiffPath** is trained using the AdamW optimizer with a linear learning rate decay starting at 1e-4, a dropout rate of 0.1, and a batch size of 64. The total number of training iterations is set to 350K. For the noise schedule, we design a square-root (sqrt) schedule to better handle the diffusion process.

Figure 6 illustrates the evolution of the Kullback-Leibler Edge Visit Divergence (KLEV) and the Jensen-Shannon Edge Visit Divergence (JSEV) as the number of training iterations increases.

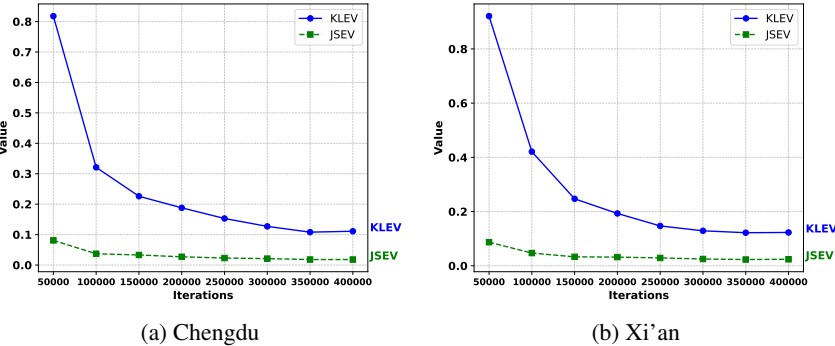

(a) Chengdu                    (b) Xi'an

Figure 6: The evolution of the KLEV and JSEV as the number of training iterations increases.

## C    ANALYSIS OF VISUAL EXPERIMENT RESULTS

### C.1    PATH LENGTH STATISTICS

We compared the frequency distribution of path lengths between the generated and real-world paths.Figure 10 presents the path length frequency distributions for both the generated and real-world paths in Chengdu and Xi'an. The visual comparison demonstrates that the length distribution of the generated paths is generally consistent with that of the real paths. However, slight deviations are observed, particularly for longer paths, where the generated paths exhibit a somewhat narrower distribution compared to the real paths. This suggests that while the model effectively generates paths of typical lengths, there may be room for improvement in generating longer, less frequent paths. Despite these minor discrepancies, the overall similarity between the length distributions indicates that the model performs well in replicating the key characteristics of the real data.

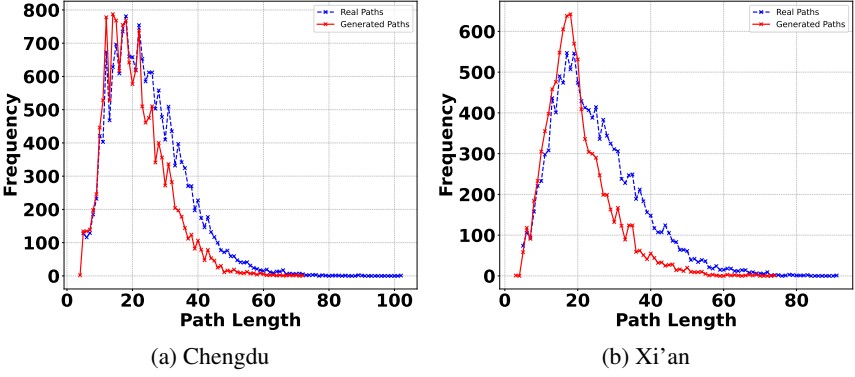

(a) Chengdu                    (b) Xi'an

Figure 7: Comparison of Real and Generated Path Length Distributions

## C.2 PATH DISTRIBUTION ADDITIONAL STATISTICS

To evaluate the effectiveness of our model in learning path distributions, we conducted additional experiments and visualized the results. Specifically, we divided the dataset's geographical area into a 3×3 grid, resulting in nine regions. We then calculated the distribution of path starting and ending points. For instance, a path starting in region 1 and ending in region 2 is recorded in cell (1, 2) of the grid. The distributions of paths generated by the real dataset and DiffPath are compared as shown below.

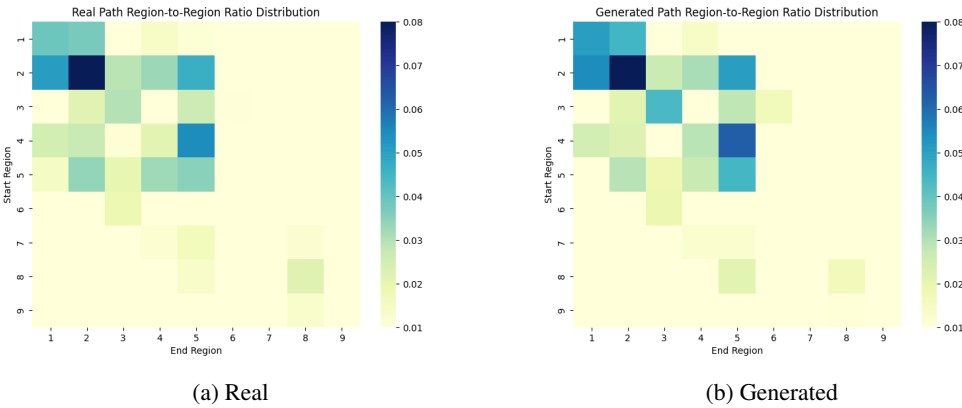

|           (a) Real           |        (b) Generated        |

Figure 8: Path distribution statistics based on origin and destination flow(Chengdu)

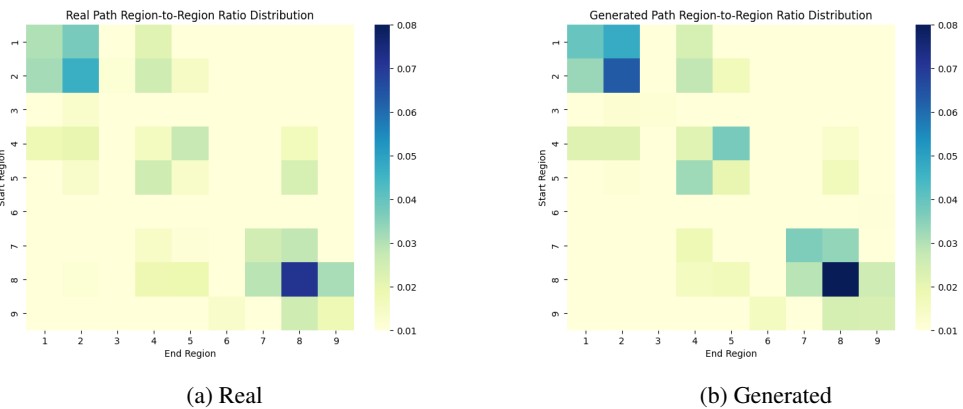

|           (a) Real           |        (b) Generated        |

Figure 9: Path distribution statistics based on origin and destination flow(Xi'an)

The visualization demonstrates that the statistical grid of paths generated by DiffPath closely resembles that of the real dataset, indicating that the overall distribution of paths generated by DiffPath aligns well with the real data. Furthermore, in combination with the results presented in Table3, we can confirm that DiffPath effectively captures the real-world path distribution.

## C.3 LOW FREQUENCY ROAD SEGMENTS STATISTICS

To assess the model's capability in capturing complex path distributions, particularly low-frequency paths, we conducted additional experiments. In these experiments, we determined the average number of road segments in the Chengdu and Xi'an datasets to be 633.90 and 473.77, respectively. Road segments that appeared fewer than 50 times in the real data were categorized as low-frequency

segments. The generation proportions of these low-frequency segments across various models are presented in the figures below.

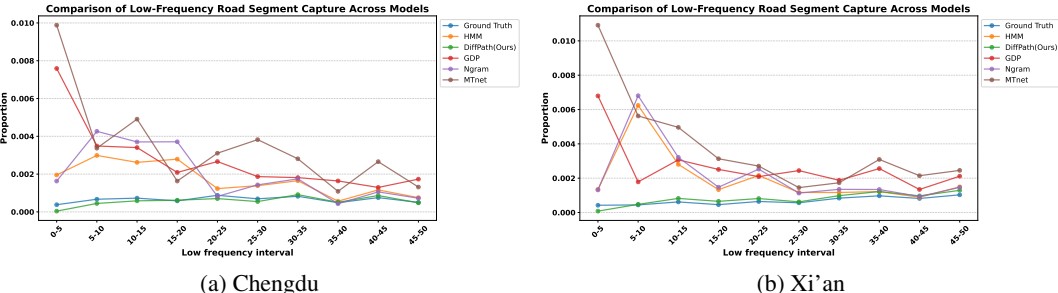

(a) Chengdu      (b) Xi'an

Figure 10: Comparison of results generated by low-frequency road segments

The results demonstrate that the generation ratios of our model align most closely with the trend line of the real dataset, indicating its superior performance in accurately capturing the distribution of low-frequency road segments within the real data.

