# OpenReview forum: "DiffPath: Generating Road Network based Path with Latent Diffusion Model"
_ICLR.cc/2025/Conference — ICLR 2025 Conference Withdrawn Submission_

### Official Review · Reviewer_Dsmz · 2024-10-17

**Soundness:** 1
**Presentation:** 2
**Contribution:** 2
**Rating:** 3
**Confidence:** 5

**Summary:**

This study model path generation using diffusion model and take advantage of transformer architecture to consider the long-term input.

**Strengths:**

1. Propose transformer-based diffusion framework for path generation and validate on real-world dataset.

**Weaknesses:**

1. The motivation of this study is not convincing. In line 59, they claimed that “”
Another significant challenge in path generation for urban road networks is… because they do not conform to most situations in reality”, if the previous model is trained based on the real-world dataset, why do these models fail to capture suck kind of reality? Besides, it is also unclear how this study addresses the claimed challenge.
2. The novelty is limited compared with the previously proposed diffusion based trajectory generation method[1,2]. The difference between this study and the previous one is only that this study adopts transformer architecture. Moreover, how do this study ensure topology constraint during path generation is not convincing. They proposed to clamp the predicted latent state to the nearest valid road segment embedding. How can generation convergence is guaranteed under this kind of operation? Besides, this operation is not theoretically guaranteed to meet the topology constraint.
3. The experimental studies are not sufficient, for example, they don’t compare with other diffusion-based trajectory generation methods [1,2].

[1] Zhu Y, Yu J J, Zhao X, et al. Controltraj: Controllable trajectory generation with topology-constrained diffusion model[C]//Proceedings of the 30th ACM SIGKDD Conference on Knowledge Discovery and Data Mining. 2024: 4676-4687.

[2] Zhu Y, Ye Y, Zhang S, et al. Difftraj: Generating gps trajectory with diffusion probabilistic model[J]. Advances in Neural Information Processing Systems, 2023, 36: 65168-65188.

**Questions:**

None

---

> ### Author Response · Authors · 2024-11-24
> **Response to Reviewer Dsmz(Part 1)**
>
> **W1**:The motivation of this study is not convincing. In line 59, they claimed that “” Another significant challenge in path generation for urban road networks is… because they do not conform to most situations in reality”, if the previous model is trained based on the real-world dataset, why do these models fail to capture suck kind of reality? Besides, it is also unclear how this study addresses the claimed challenge.
>
> **A1**:Thank you for your valuable feedback. Due to space limitations, we did not elaborate in the paper on the limitations of previous models. Here, we provide an analysis of three representative models. It is important to clarify that we are referring specifically to path generation models, not trajectory generation models.
>
> First, we consider two count-based models, N-gram and HMM. These models estimate the transition probabilities between consecutive road segments purely based on observed counts. In other words, they focus solely on local connections between adjacent nodes, lacking the ability to capture the influence of non-adjacent nodes.
> The third model, MTNet, combines recurrent neural networks with meta-learning techniques to update the next node's information based on historical states. However, MTNet also places excessive emphasis on the weights of adjacent nodes, while the influence of global information diminishes as the path length increases. As a result, it fails to effectively capture the broader global structure of the path. This limitation is further validated by our experimental results, which demonstrate the superiority of our proposed method in addressing these challenges.
>
> The proposed framework addresses the identified challenges by leveraging the inherent strengths of the Transformer architecture and diffusion modeling, while incorporating task-specific enhancements tailored for path generation. Transformers excel at capturing long-range dependencies, which is crucial for maintaining the global coherence of paths, while the diffusion process enables effective sampling and refinement to ensure both diversity and realism.
>
> Building on these advantages, our framework introduces specific adaptations to meet the unique demands of path generation. We integrate positional embeddings and a clamping mechanism during the reverse diffusion process, ensuring that generated paths adhere to topological constraints and maintain contextual consistency. Additionally, our custom loss function mitigates the challenges posed by the long-tail distribution of road sections, enabling the model to better learn from underrepresented yet structurally significant regions.

---

> ### Author Response · Authors · 2024-11-24
> **Response to Reviewer Dsmz(Part 2)**
>
> **W2**:The novelty is limited compared with the previously proposed diffusion based trajectory generation method[1,2]. The difference between this study and the previous one is only that this study adopts transformer architecture. Moreover, how do this study ensure topology constraint during path generation is not convincing. They proposed to clamp the predicted latent state to the nearest valid road segment embedding. How can generation convergence is guaranteed under this kind of operation? Besides, this operation is not theoretically guaranteed to meet the topology constraint.
>
> **A2**:Thank you for your valuable feedback. We would like to clarify the key differences and novel contributions of our approach. Unlike previous studies, our method goes beyond simply adopting a transformer architecture. We introduced targeted adaptations to address the specific challenges of path generation, including the use of positional embeddings tailored to road network topology and a clamping mechanism within the diffusion process.
>
> The clamping mechanism is designed to enforce road network constraints and enhance model convergence. However, we intentionally choose not to apply strong constraints, such as incorporating road network graphs in the loss function to assess the continuity of generated paths. We find that even without these strong constraints, the generated paths exhibit only marginal differences in continuity. Furthermore, we believe that adding such constraints would unnecessarily limit the model's generative capacity.
>
> The transformer backbone is utilized to capture long-range dependencies and global context, particularly in situations where local constraints alone might lead to unrealistic path decisions. This integration of both global and local information is key to generating paths that are both valid and realistic, as demonstrated by our evaluation metrics and qualitative analysis.
>
> Finally, we have introduced novel evaluation metrics and region-based analyses to rigorously validate the generated paths. These metrics not only assess the realism of the paths but also highlight the model’s capability to handle complex road networks and diverse travel patterns, further distinguishing our work.
>
> **W3**:The experimental studies are not sufficient, for example, they don’t compare with other diffusion-based trajectory generation methods [1,2].
>
> **A3**:Thanks to the reviewer's advice, we take DiffTraj into consideration when selecting the baseline for the trajectory generation task.   Since our task is fundamentally path generation, DiffTraj is not initially selected as a baseline.   Subsequently, we adapt our dataset to the DiffTraj model as suggested by the reviewer.   Due to the limitations of the dataset, we are making more attempts and will release the results as soon as possible.

---

### Official Review · Reviewer_haW5 · 2024-10-29

**Soundness:** 3
**Presentation:** 2
**Contribution:** 2
**Rating:** 5
**Confidence:** 4

**Summary:**

This paper presents DiffPath to address the challenges of complex segment distribution in path generation and to ensure global consistency of the generated paths. Experimental results validate its effectiveness in generating realistic paths.

**Strengths:**

S1. The solution to the path generation problem offers a certain degree of protection for personal privacy.
S2. This paper is the first to attempt the use of latent diffusion models, which excel in generative tasks, in the context of path generation, along with targeted design considerations.

**Weaknesses:**

W1. Compared to the de-identification of real path data, the issues of accuracy and computational complexity in path generation appear more complex and unreliable.
W2. In related studies, the assumption of maintaining symmetry in the adjacency matrix of existing diffusion models may inaccurately represent one-way streets as bidirectional. This warrants a more in-depth discussion, as directed graphs do not necessarily require a symmetric structure in their adjacency matrices.
W3. The legend does not correspond with the paper's description; please verify the relationship between paths P1 and P2 in Figure 2 and the accuracy of the related statement in line 64.
W4. The ablation study analyzes replacing the Transformer with UNet but lacks a thorough analysis of the Diffusion module.
W5. No reproducible code is provided, making it impossible to verify the validity of the research findings.

**Questions:**

Q1. Due to the errors in the legend and related descriptions, I do not understand why "P2 does not consider that selecting $v_4$ will result in a longer path to reach $v_7$." Is the distance from $v_2$ to $v_7$ indeed longer? More justification is needed to demonstrate that the generated path adheres to the constraints of the road network to substantiate this challenge.
Q2. Diffusion-based models typically exhibit high complexity; how does the computational complexity of DiffPath compare to the baseline?

---

> ### Author Response · Authors · 2024-11-24
> **Response to Reviewer haW5 (Part 1)**
>
> **W1**:Compared to the de-identification of real path data, the issues of accuracy and computational complexity in path generation appear more complex and unreliable.
>
> **A1**:We sincerely appreciate the reviewer’s comment. The collection of real-world data is inherently limited and often incurs high costs for large-scale acquisition. In contrast, path generation enables the creation of large volumes of data, offering a more scalable and cost-effective alternative. Furthermore, we plan to explore controlled generation techniques and city-to-city transferable path generation in future research to make the process more efficient and reliable.
>
> **W2**:In related studies, the assumption of maintaining symmetry in the adjacency matrix of existing diffusion models may inaccurately represent one-way streets as bidirectional. This warrants a more in-depth discussion, as directed graphs do not necessarily require a symmetric structure in their adjacency matrices.
>
> **A2**:We appreciate the reviewer for pointing out this critical limitation regarding the assumption of symmetry in adjacency matrices, particularly in the context of one-way streets. In our original experiments, we strictly followed the experimental setup of the baseline studies, including their design choice to use symmetric adjacency matrices, to ensure a fair comparison. However, based on the reviewer’s valuable suggestion, we conducted additional experiments that adapt the matrix design to reflect the asymmetry of directed graphs. Specifically, we modified the adjacency matrix to encode the directionality of road segments, allowing the model to better account for one-way streets and other directional constraints.Using the digraph matrix, you can see that the GDP model similarity score becomes higher because fewer discontinuous reverse paths are generated. However, the KLEV index is significantly worse, perhaps because the construction of the directed graph restricts the generation of paths, or the modeling assumptions of the baseline approach may inherently limit its ability to effectively utilize directional information. And we modify our analysis in the experimental analysis section.
> | **City**       | **Metrics** | **N-gram** | **HMM** | **GDP** | **GDP_Digraph** | **MTnet** | **DiffPath (Ours)** |
> |-----------------|-------------|------------|---------|---------|----------------|-----------|---------------------|
> |                 | SS          | 0.701      | 0.681   | 0.616   |    0.765       | 0.821     | **0.933**           |
> | **Chengdu**     | KLEV        | 0.140      | 0.135   | 0.686   |    0.774       | 0.129     | **0.106**           |
> |                 | JSEV        | 0.033      | 0.028   | 0.159   |    0.120       | 0.038     | **0.018**           |
> |-----------------|-------------|------------|---------|---------|----------------|-----------|---------------------|
> |                 | SS          | 0.628      | 0.633   | 0.571   |    0.793       | 0.772     | **0.893**           |
> | **Xi'an**       | KLEV        | 0.133      | 0.130   | 0.697   |    0.818       | 0.127     | **0.122**           |
> |                 | JSEV        | 0.031      | 0.025   | 0.147   |    0.134       | 0.033     | **0.023**           |

---

> ### Author Response · Authors · 2024-11-24
> **Response to Reviewer haW5 (Part 2)**
>
> **W3**. The legend does not correspond with the paper's description; please verify the relationship between paths P1 and P2 in Figure 2 and the accuracy of the related statement in line 64.
>
> **A3**:Thank you for bringing this issue to our attention. We have carefully reviewed Figure 2 and the accompanying description on line 64 and identified the inconsistency. The legend in the original figure did not accurately reflect the relationship between paths P1 and P2, leading to a mismatch with the description in the paper. We have revised the figure to ensure that the legend corresponds correctly to the paths and updated the statement on line 64 for accuracy. The corrected figure and statement will be included in the revised version in line 64.
>
> **W4**:The ablation study analyzes replacing the Transformer with UNet but lacks a thorough analysis of the Diffusion module.
>
> **A4**:Thank you for highlighting this point. While our current ablation study focuses on comparing Transformer and UNet architectures, we recognize the importance of analyzing the Diffusion module. We are actively conducting experiments.  These results will be included in future updates to provide a more comprehensive analysis.
>
> **W5**:No reproducible code is provided, making it impossible to verify the validity of the research findings.
>
> **A5**:Thank you for your feedback. We understand the importance of providing reproducible code to validate research findings. While the code is not publicly available at this stage, we are committed to releasing it in the near future to ensure transparency and reproducibility. We appreciate your understanding and patience as we finalize the necessary preparations for its release.

---

> ### Author Response · Authors · 2024-11-24
> **Response to Reviewer haW5 (Part 3)**
>
> **Q1**. Due to the errors in the legend and related descriptions, I do not understand why "P2 does not consider that selecting v4 will result in a longer path to reach v7." Is the distance from v2 to v7 indeed longer? More justification is needed to demonstrate that the generated path adheres to the constraints of the road network to substantiate this challenge.
>
> **A1**:Thank you for your valuable feedback. We have corrected the inaccuracies in the legend and descriptions related to Figure 2. The revised version ensures the alignment of the legend and statements, making the example clearer.The purpose of the example in Figure 2 is to illustrate a key challenge in path generation. Specifically, P1 represents a plausible real-world path that people might take to travel from v1 to v7. In contrast, P2 demonstrates a scenario where the lack of global context leads the model to make suboptimal choices at intermediate steps. For example, when the model reaches v2, it may select v4 due to local constraints or road network connectivity, resulting in a longer and less realistic path to v7. This behavior highlights the difficulty in generating paths that not only satisfy road network constraints but also align with realistic travel patterns. The challenge we aim to address is that the model, without sufficient global consideration, may generate paths like P2, which are rare and impractical in real-world scenarios when traveling from v1 to v7. We hope this clarification captures the intended explanation and provides a stronger justification for the challenges our work addresses. Thank you for bringing this to our attention, as it allows us to refine the manuscript further.
>
> **Q2**:Diffusion-based models typically exhibit high complexity; how does the computational complexity of DiffPath compare to the baseline?
>
> **A2**:DiffPath has a higher computational complexity compared to the baseline approach, mainly due to the iterative diffusion process and Transformer architecture.These elements are essential for capturing global coherence and local transitions, enabling the generation of realistic and contextually accurate paths.Although DiffPath is more computationally demanding, its ability to generate diverse and high-fidelity paths justifies the trade-off, particularly for applications requiring realistic path generation.

---

> > ### Comment · Reviewer_haW5 · 2024-11-26
> >
> > Thank the authors for their response.  Although the authors have acknowledged and corrected the corresponding errors, the existence of these fundamental issues still affects the rigor of the paper. Furthermore, the response still lacks a complexity analysis and verifiable code. As the authors mentioned, there are many areas in this work that could be further improved.

---

### Official Review · Reviewer_CWAT · 2024-11-02

**Soundness:** 2
**Presentation:** 3
**Contribution:** 2
**Rating:** 5
**Confidence:** 3

**Summary:**

This paper introduces DiffPath, a framework aimed at addressing path generation using a latent diffusion model combined with a transformer. The authors highlight two key challenges in prior work on path generation: complex path distributions and ensuring global coherence in generated paths. They suggest that these issues can be addressed through the integration of latent diffusion models with a transformer architecture. The experimental results indicate that DiffPath performs well on two real-world datasets.

**Strengths:**

1. The methodology is straightforward and easy to follow.
2. The writing is clear and accessible.
3. The framework has good performance on real-world datasets.

**Weaknesses:**

1. The core contribution is confusing. This work seems to simply apply the diffusion transformer model on the path generation task without additional optimization specific to this task.
2. While the authors claim that the proposed model addresses the challenges of capturing complex path distributions and ensuring coherence in generated paths, there is a lack of experimental evidence and analysis to support these claims.

**Questions:**

1. What is the core contribution of this work?
2. How does the proposed framework tackle the claimed challenges?

---

> ### Author Response · Authors · 2024-11-24
> **Response to Reviewer CWAT (Part 1)**
>
> **W1**:The core contribution is confusing. This work seems to simply apply the diffusion transformer model on the path generation task without additional optimization specific to this task.
>
> **A1**:We sincerely appreciate the reviewer’s comment and would like to clarify the core contributions of our work. While the diffusion transformer framework forms the foundation of our approach, our contributions involve substantial task-specific innovations tailored to the challenges of path generation. To address the long-tail distribution of road segments, we introduce a custom loss function within the diffusion process, enabling the model to effectively learn the structural nuances of the data and generate diverse, realistic paths, particularly in underrepresented regions. Additionally, we adapt the transformer-based architecture by incorporating positional embeddings and a clamping mechanism during the reverse diffusion process, ensuring that generated paths adhere to road network constraints while maintaining contextual relevance.
> To rigorously evaluate the realism of the generated paths, we propose a novel similarity score that measures both local transitions and global coherence against real-world trajectories. This is further supported by complementary quantitative metrics, such as KLEV and JSEV, along with comprehensive visualizations and region-to-region transition analyses, which collectively demonstrate the model’s ability to capture the complexity and diversity of real-world travel patterns. These contributions, while leveraging the strengths of the diffusion transformer framework, represent significant methodological advancements specifically designed to address the unique challenges of path generation.
>
> **W2**:While the authors claim that the proposed model addresses the challenges of capturing complex path distributions and ensuring coherence in generated paths, there is a lack of experimental evidence and analysis to support these claims.
>
> **A2**:
> We appreciate the reviewer’s concern and have conducted additional experiments to further validate our claims. Specifically, we performed region-to-region flow statistics by dividing the study area into a 3×3 grid and analyzing the distribution of paths based on their starting and ending regions. For example, a path starting in Region 1 and ending in Region 2 is recorded in cell (1,2) of the flow matrix. This analysis provides insights into how well the generated paths adhere to real-world spatial patterns.
>
> The results of these experiments are shown in Figure 9 in our updated paper. Combined with the metrics presented in our paper, KLEV and JSEV, which evaluate the segment-wise distribution of intermediate road segments, these region-to-region flow statistics offer additional validation of the model's capability to capture complex path distributions. Together, these analyses demonstrate that our model not only generates paths consistent with the real-world road network but also effectively maintains coherence and diversity across different spatial regions. Additional experiments have been added in the appendix of the paper.

---

> ### Author Response · Authors · 2024-11-24
> **Response to Reviewer CWAT (Part 2)**
>
> **Q1**:What is the core contribution of this work?
>
> **A1**:The core contribution of this work lies in advancing the path generation domain by integrating state-of-the-art Transformer and diffusion modeling with task-specific innovations that address the unique challenges of generating realistic and coherent paths. Rather than a straightforward application of these models, we developed a comprehensive framework tailored specifically for path generation, effectively balancing long-range dependencies and local transitions while addressing the inherent complexities of this task.
>
> Key contributions include the introduction of a custom loss function designed to mitigate the long-tail distribution of road segments. This ensures that the generated paths are diverse and representative, particularly in underrepresented areas of the road network. Additionally, we adapted the diffusion process with novel positional embeddings and a clamping mechanism to guarantee topological validity and contextual coherence, both of which are essential for replicating real-world travel patterns.
>
> To complement these methodological innovations, we proposed a new similarity score tailored for evaluating path realism. This metric rigorously assesses the generated paths by considering both local transitions and global coherence, providing a nuanced understanding of how closely the generated paths align with real-world trajectories.
>
> Through the combination of advanced modeling techniques, domain-specific optimizations, and innovative evaluation metrics, our work makes a significant contribution to generating paths that not only replicate the structural and contextual intricacies of real-world travel patterns but also set a new benchmark for fidelity and diversity in the field.
>
> **Q2**:How does the proposed framework tackle the claimed challenges?
>
> **A2**:The proposed framework tackles the claimed challenges by leveraging the inherent strengths of the Transformer architecture and diffusion modeling while incorporating task-specific enhancements tailored for path generation. Transformers excel at capturing long-range dependencies, which is crucial for maintaining the global coherence of paths, while the diffusion process enables effective sampling and refinement of paths to ensure diversity and realism.
>
> Building on these advantages, our framework introduces specific adaptations to address the unique demands of path generation. We integrate positional embeddings and a clamping mechanism during the reverse diffusion process, ensuring that generated paths adhere to topological constraints and maintain contextual consistency. Furthermore, our custom loss function mitigates the challenges posed by the long-tail distribution of road sections, enabling the model to better learn underrepresented yet structurally significant regions. These targeted innovations amplify the capacity of the Transformer and diffusion framework to handle the complexity of path generation and replicate realistic travel patterns, making it a robust solution to the challenges identified.

---

### Official Review · Reviewer_oQM3 · 2024-11-05

**Soundness:** 2
**Presentation:** 3
**Contribution:** 2
**Rating:** 5
**Confidence:** 4

**Summary:**

This paper introduces DiffPath, a path generation model that uses a latent diffusion model (LDM) and a transformer to generate realistic synthetic road paths, addressing privacy concerns and data limitations in urban navigation and planning. DiffPath embeds discrete paths into a continuous latent space, allowing it to capture complex path distributions and ensuring coherence between adjacent and distant road segments. By incorporating a customized loss function, the model aims to generate paths with rare segments often missed by traditional methods. Experimental results on datasets from Chengdu and Xi’an show that DiffPath outperforms existing approaches in generating synthetic paths that align well with real-world road networks.

**Strengths:**

- This paper tackles a practical problem in the urban computing scenario. It aims to address privacy concerns and data limitations in urban navigation and planning, which is of high practical value.

- The paper proposes a unique angle that is overlooked in previous works. They tend to focus on the local smoothness of the path but lose global-level constraints.

- The paper is well-written and easy to follow.

**Weaknesses:**

- The experiments conducted are not enough to evaluate the claimed advantages, i.e., generate more realistic paths, especially those low-frequency ones.

- The proposed method is rather straightforward. Moreover, I think using the transformer and diffusion modeling instead of autoregressive modeling are both vital for capturing long-range correlation within a path.

- Similarity matric seems to suffer from bias issues. What if the generated paths are all the same but highly similar to one ground truth?

**Questions:**

Please see my review above.

---

> ### Author Response · Authors · 2024-11-24
> **Response to Reviewer oQM3 (Part 1)**
>
> **W1**:The experiments conducted are not enough to evaluate the claimed advantages, i.e., generate more realistic paths, especially those low-frequency ones.
>
> **A1**：Thanks for your suggestions. To address the concern regarding our model's ability to generate realistic paths, particularly those involving low-frequency road segments, we conduct additional experiments specifically targeting this aspect. First, we statistically identify low-frequency road segments within the real datasets. Subsequently, we analyze the generation proportion of these low-frequency road segments across different models, including our proposed framework, to evaluate how accurately each model captures these underrepresented segments relative to their true distribution in the real data. The results demonstrate that our model consistently achieves a closer alignment with the real data compared to baseline models in generating low-frequency road segments, of which the figures are shown in Figure 10 in our updated paper. These experiments are conducted on two diverse datasets, Chengdu and Xi’an, further validating the robustness and generalizability of our conclusions. Additional experiments are included in the appendix of the paper.
>
> **W2**:The proposed method is rather straightforward. Moreover, I think using the transformer and diffusion modeling instead of autoregressive modeling are both vital for capturing long-range correlation within a path.
>
> **A2**:We appreciate your recognition of the potential of Transformers and diffusion models in our proposed method. Indeed, the inherent strengths of these frameworks, such as the ability of Transformers to model long-range correlations and the capacity of diffusion models to capture complex data distributions, are critical for tackling the challenges in path generation. However, applying these frameworks to the domain of discrete path generation in road networks involves unique complexities that we have specifically addressed.
>
> To this end, we developed a latent diffusion framework tailored for sequential path data, embedding discrete paths into a continuous latent space. This approach enables iterative denoising and reconstruction of paths while adhering to road network constraints. Furthermore, to address the long-tail distribution of road segments, we integrated a custom loss function into the diffusion process. This enhancement enables the model to better capture the structural nuances and distribution of the original data, facilitating the generation of diverse and realistic paths, particularly in urban centers and underrepresented regions.
>
> Additionally, we adapted the Transformer architecture to the path generation task by incorporating positional embeddings and a clamping mechanism during the reverse diffusion process. These adaptations ensure that the generated paths maintain both topological validity and contextual awareness, key attributes for replicating realistic travel patterns.
>
> While the underlying frameworks of Transformers and diffusion models provide a strong foundation, our task-specific adaptations and methodological innovations substantiate the significant contributions of this work. We are grateful for the reviewer’s feedback and welcome further suggestions for improvement or clarification.

---

> ### Author Response · Authors · 2024-11-24
> **Response to Reviewer oQM3 (Part 2)**
>
> **W3**:Similarity matric seems to suffer from bias issues. What if the generated paths are all the same but highly similar to one ground truth?
>
> **A3**:Thanks for your comments. We acknowledge that solely relying on similarity metrics may indeed lead to biased evaluations. To address it, we incorporated two additional distributional metrics, KLEV and JSEV, in our paper to provide a more comprehensive assessment of the generated paths. By jointly considering these three metrics, we achieve a holistic evaluation of the path generation quality. Specifically, the similarity metric confirms that the generated paths closely match the ground truth, and the distributional metrics further validate that the generated paths are diverse and effectively capture the broader characteristics of real-world trajectory distributions.Combining these metrics ensure a more balanced and reliable evaluation.

---

### Author Response · Authors · 2024-11-24
**Summary of Changes**

Thanks to all reviewers. We have received many constructive comments and suggestions. Based on these we have revised our paper. All updates are highlighted in blue in the revised paper and the main updates are summarized as follows:

In the introduction, we have rewritten the relevant descriptions and contributions in Figure 1 to make them clearer.

We changed our analysis of the GDP model to more fully assess the effects of the baseline model.

We add capture validation experiments on low frequency paths, visualizing the results in Appendix C.

We have added visualizations of the path distribution, as detailed in Appendix C.

---

### Note · Authors · 2025-01-16

I have read and agree with the venue's withdrawal policy on behalf of myself and my co-authors.